# A systematic review and narrative synthesis of the research provisions under the Mental Capacity Act (2005) in England and Wales: Recruitment of adults with capacity and communication difficulties

Oluseyi Florence Jimoh[1]*, Hayley Ryan[2], Anne Killett[1], Ciara Shiggins[3], Peter E. Langdon[4], Rob Heywood[5], Karen Bunning[1]

1 School of Health Sciences, University of East Anglia, Norwich Research Park, Norwich, United Kingdom, 2 Norwich Medical School, University of East Anglia, Norwich Research Park, Norwich, United Kingdom, 3 Centre of Research Excellence in Aphasia Recovery and Rehabilitation, La Trobe University, Melbourne Victoria, Australia, 4 Centre for Educational Development, Appraisal and Research, New Education Building, Westwood Campus, University of Warwick, Coventry, United Kingdom, 5 School of Law, University of East Anglia, Norwich Research Park, Norwich, United Kingdom

* O.Jimoh@uea.ac.uk

## Abstract

### Background

The Mental Capacity Act (MCA, 2005) and its accompanying Code of Practice (2007), govern research participation for adults with capacity and communication difficulties in England and Wales. We conducted a systematic review and narrative synthesis to investigate the application of these provisions from 2007 to 2019.

### Methods and findings

We included studies with mental capacity in their criteria, involving participants aged 16 years and above, with capacity-affecting conditions and conducted in England and Wales after the implementation of the MCA. Clinical trials of medicines were excluded. We searched seven databases: Academic Search Complete, ASSIA, MEDLINE, CINAHL, PsycArticles, PsycINFO and Science Direct. We used narrative synthesis to report our results. Our review follows Preferred Reporting Items for Systematic Reviews and is registered on PROSPERO, CRD42020195652.

28 studies of various research designs met our eligibility criteria: 14 (50.0%) were quantitative, 12 (42.9%) qualitative and 2 (7.1%) mixed methods. Included participants were adults with intellectual disabilities (n = 12), dementia (n = 9), mental health disorders (n = 2), autism (n = 3) and aphasia after stroke (n = 2). We found no studies involving adults with acquired brain injury. Diverse strategies were used in the recruitment of adults with capacity and communication difficulties with seven studies excluding individuals deemed to lack capacity.

**Data Availability Statement:** All relevant data are within the manuscript and its Supporting information files.

**Funding:** KB OSAP/43239 NUFFIELD Foundation https://www.nuffieldfoundation.org/ The funders had no role in study design, data collection and analysis, decision to publish, or preparation of the manuscript.

**Competing interests:** The authors have declared that no competing interests exist.

## Conclusions

We found relatively few studies including adults with capacity and communication difficulties with existing regulations interpreted variably. Limited use of consultees and exclusions on the basis of capacity and communication difficulties indicate that this group continue to be under-represented in research. If health and social interventions are to be effective for this population, they need to be included in primary research. The use of strategic adaptations and accommodations during the recruitment process, may serve to support their inclusion.

## Introduction

Informed consent is a requirement of intrusive research [1], which upholds the principles of autonomous decision-making with provisions for the protection of those who lack capacity [2,3]. It requires that the person can understand and retain relevant information, weigh up the implications of participation, and communicate a decision [4–6]. However, our society also includes people who lack mental capacity and people with communication difficulties, either as separate impairments or in combination, referred to in this review as adults with capacity and communication difficulties (CCDs). The number of people affected by such difficulties is rising and include people with dementia [7], stroke [8], acquired brain injury [9], mental health difficulties [10], autism and intellectual disabilities [11,12]. In the context of a rising prevalence of people living CCD, there is a need for research to advance our understanding of these conditions and to improve evidence-based interventions. However, research shows that people living with CCDs continue to be under-represented in research [13,14].

In England and Wales, the Mental Capacity Act (MCA) (2005) [2] and its accompanying Code of Practice (CoP) [2,15] were originally introduced to protect the rights of adults who may lack capacity for autonomous decision-making in relation to treatment, welfare and finance. There are separate provisions for research (CoP: Chapter 11). Different legislation is provided in other countries of the UK: the Adults with Incapacity (Scotland) Act 2000 (AWIA); the Mental Capacity Act (Northern Ireland) (2016). In Ireland, it is the Assisted Decision Making (Capacity) Act 2015. However, the current review pertains to the Mental Capacity Act (2005) in England and Wales. The MCA applies to 'intrusive' research, which refers to research that would require consent if it were conducted on persons with mental capacity [2]. It does not apply to clinical trials of medicines which is governed by different legislation (The Medicines for Human Use Clinical Trials Regulations) [16].

For the purposes of research, there is the presumption of capacity unless there is a reason to believe that a person lacks capacity (CoP 2007). Before deciding that someone lacks capacity, the CoP (2007) recommends the provision of relevant information, communicated in the most appropriate way [15]. Whilst practical details are not given, there is general encouragement for presenting project information to suit the processing capabilities of potential participants. For example, support for the person's understanding of what research participation entails might include: information sheets rendered in simple language with or without pictorial support; a simulated data collection procedure shown on video; questions and answer opportunities in conversations about a project; and use of manual sign and gesture to augment meanings [15,17–19]. Relevance theory [20] argues that people find it easier to engage with and understand information that is most relevant to them and requires the least cognitive

effort. The form of the message interacts with the person's cognitive abilities, prior experience and underlying knowledge. On this latter point, the person's familiarity with the subject matter contributes to their perception of possible cognitive gain, which in turn optimises the potential relevance of information to them [20]. This asserts the importance of addressing the information-processing needs of the target population for successful recruitment to studies, particularly where CCDs are present.

Notwithstanding the presumption of capacity [CoP: 11.4; MCA S.1(2)], an assessment of an individual's capacity is a requirement [2,15] when concerns are raised about capacity. For this purpose, a two-stage test is recommended [MCA S.3; CoP 4:10]. There is no one standard method for the purpose, with many researchers using locally-developed initiatives [21,22]. Capacity is defined as time and decision-specific, variable according to complexity of information [23], and possibly fluctuating over time [24]. The distinction between capacity and lack of capacity is far from straightforward [24,25]. Furthermore, the presence of communication and cognitive impairments may complicate the informed consent process [26–30] by masking true competence in people with, for example, early stage dementia, moderate intellectual disability [31,32], aphasia following stroke [33,34] and autistic spectrum disorder [35]. To circumnavigate some of these difficulties, researchers have developed person-centred approaches [24] characterised by flexibility and support from family and friends [36].

A proven lack of capacity requires the advice of a consultee, either personal (e.g. relatives, friends, unpaid carer) or nominated (e.g. healthcare professionals) [36,37], about the individual's likely wishes and feelings concerning research participation (CoP: 11.20) [2]. In the context of a consultee's affirmative advice, researchers are required to prioritise the interest of the participant above that of science and the society (CoP: 11.20; CoP 11.29), considering their wishes and feelings throughout the research process (CoP 11.29) [15]. In such cases, expressions of: assent (a person's 'permission or affirmative agreement to something) [38]; and dissent (a person's disagreement or refusal), are recognised appropriately [36]. This aligns with the principle of partial participation [39], which acknowledges that gradations of involvement are possible. Gatekeepers such as residential home managers, carers and health professionals, are uniquely placed to facilitate access to those with CCD because of an existing relationship with the person [40]. Thus, the individual's participation in research is not only dependent on autonomous decision-making or consultee advice, but upon overcoming additional barriers such as permission from gatekeepers.

There has been limited consideration of intrusive research under the MCA [41,42]. Previous reviews have focused on MCA provisions in relation to health and social care practice [22,43] and clinical trials of medicines, which is governed by different legislation (The Medicines for Human Use Clinical Trials Regulations [16]. Provisions for intrusive research under the MCA have been criticised for a lack of clarity leading to variable interpretations [21,44,45]. Considering these challenges, the aim of this systematic review was to develop an understanding of how adults with CDD have been included and accommodated within research studies within England and Wales following the implementation of the MCA, 2005.

## Methods

This systematic review of the literature was carried out following PRISMA guidance [46]. The review protocol (See S1 File) was prospectively registered in Prospero with Registration number CRD42020195652 [47]. In the protocol, we used the term "adults with impairments of capacity and/or communication (ICC)". This has been refined and modified through our interactions with our stakeholders to "adults with capacity and communication difficulties".

## Search strategy and eligibility criteria

We included studies conducted in England and/or Wales from 2007 (the year the Mental Capacity Act 2005, was implemented; CoP: DfCA, 2007) to 2019. The search framework focused on adults with CCD and the MCA (2005). Multiple terms, representative of the primary stakeholder groups (i.e., autism; aphasia; dementia; head injury (OR brain injury); learning disability (OR intellectual disability), were used in combination with (AND) mental capacity (OR) informed consent and applied to the following databases: Academic Search Complete, ASSIA, MEDLINE, CINAHL, PsycArticles, PsycINFO and Science Direct. The initial search strategy was developed in MEDLINE and adjusted according to the indexing systems of other databases (See S2 File). The first search was carried out on 11[th] December 2019 and an updated search on 13[th] July 2020, to identity any additional papers.

## Study selection

Search results were combined into a single Endnote file, citations were screened, and duplicates removed in accordance with the PRISMA statement [46]. Two researchers (FJ and HR) then independently screened all titles to identify relevant studies according to the eligibility criteria (Table 1). Then, abstracts were reviewed to identify studies to undergo full-text review. Disagreements were resolved by discussion between the two researchers. We did not search grey literature sources but supplemented searches with backwards and forward searches of the references listed in the included studies.

## Data extraction and quality assessment

The review set out to identify, describe and synthesise the procedures and accommodations used by researchers to support the inclusion and participation of adults with impairments of capacity and communication in research. The data extraction table was therefore designed to

**Table 1. Eligibility criteria.**

|  | Inclusion Criteria | Exclusion Criteria |
|---|---|---|
| Population | • Studies conducted in England and/or Wales from 2007, when the Mental Capacity Act (2005) was implemented.<br>• Participants aged 16 years and above (the age at which the MCA applies), with communication and/or capacity difficulties (e.g. associated with autism; stroke; mental health; dementia; acquired brain injury; and intellectual disabilities); | • Research studies governed by The Medicines for Human Use (Clinical Trials) Regulations 2004.<br>• Research using tissue samples.<br>• Secondary data. |
| Intervention | • Invoking the provisions for research under the MCA (2005). | |
| Outcomes | • Demographic data<br>• Recruitment procedures<br>• Accommodations supporting research participation. | |
| Study designs | Any; quantitative, qualitative, mixed study design | |
| Publication types | *Primary empirical studies from peer-reviewed literature | |
| Publication year | 2007 to 2019 | |
| Language | English language | |

Notes:

*The year the study was conducted indicated when participants were recruited.

When the date was not provided, clarification was sought by sending an email to the corresponding author and searching the publicly available Health Research Authority (HRA) database. Finally, where this could not be established, we back-tracked three years from publication data on the basis that the majority of studies are published within 30 months post the live period of a study (i.e., from 2010) [48].

capture this information and is presented in the supplementary material (S1 Table). Two researchers (FJ and HR) extracted data independently using a Microsoft Excel-based broad extraction sheet, which detailed: population-type by diagnosis, inclusion/exclusion criteria, sample size, sampling method, information format, capacity assessment procedure, informed consent procedure, research accommodations, consultee involvement, use of gatekeepers and the year of study. Data were summarised and a third researcher KB reviewed and confirmed the data extraction.

The Mixed Methods Appraisal Tool (MMAT) [49], for concurrent critical appraisal of quantitative, qualitative and mixed-methods primary research was applied [50]. The MMAT has established content validity, it has been piloted across all methodologies; quantitative, qualitative and mixed methods research designs [50,51]. Compared with other tools, the MMAT includes specific criteria for appraising mixed methods studies. While critical appraisal tools are more widely available for quantitative and qualitative research, there has not been consensus on quality criteria for mixed methods research [52].

The tool results in a methodological rating of between one and five (with five being the highest quality), for each study, based on the evaluation of study selection bias, study design, data collection methods, sample size, intervention integrity, and analysis. An overall quality score and a descriptive summary was derived for each study [49]. A score of 4–5 indicated a 'high quality'; 3 indicated 'moderate'; 2 or less indicated 'low quality'. For mixed-method studies, each methodological element was assessed separately, and the lowest quality score included. A second researcher (KB) independently checked the reliability of the quality assessment on a random sample of studies (17%) [53], with perfect agreement (k = 1.0) [54]. As the review is exploratory, no study was excluded based on quality assessment since they may still provide valuable insight [53].

## Data analysis

To account for methodological diversity and sample variability, we employed narrative synthesis in the report of results [55,56]. Using a textual approach, a descriptive summary of the included studies focused on the recorded fields in the broad extraction sheet and the relationships within and between the studies examined.

# Results

## Search results

Search results are summarised in the Preferred Reporting Items for Systematic Reviews and Meta-analyses (PRISMA) flowchart (Fig 1 and S2 Table) [46].

Our initial search identified 2116 studies and a repeat search identified a further 614 studies. Following removal of duplicates, screening and full textual review of 126 studies, of which 20 met the inclusion criteria. A further 8 studies were identified after reference and citation searches.

## Characteristics of included studies

The key characteristics of the included studies are presented in supplementary S3 Table. Included participants were said to have intellectual disabilities (n = 12; 42.9%); dementia (n = 9; 32.1%); autism spectrum disorders (n = 3; 10.7%); mental health disorders (n = 2; 7.1%); and aphasia after stroke (n = 2; 7.1%). None were said to have brain injury. Study designs included quantitative (n = 14; 50.0%); qualitative (n = 12; 42.9%) and mixed methods (n = 2; 7.1%). Samples were drawn mainly from hospital in-patients or attending outpatient

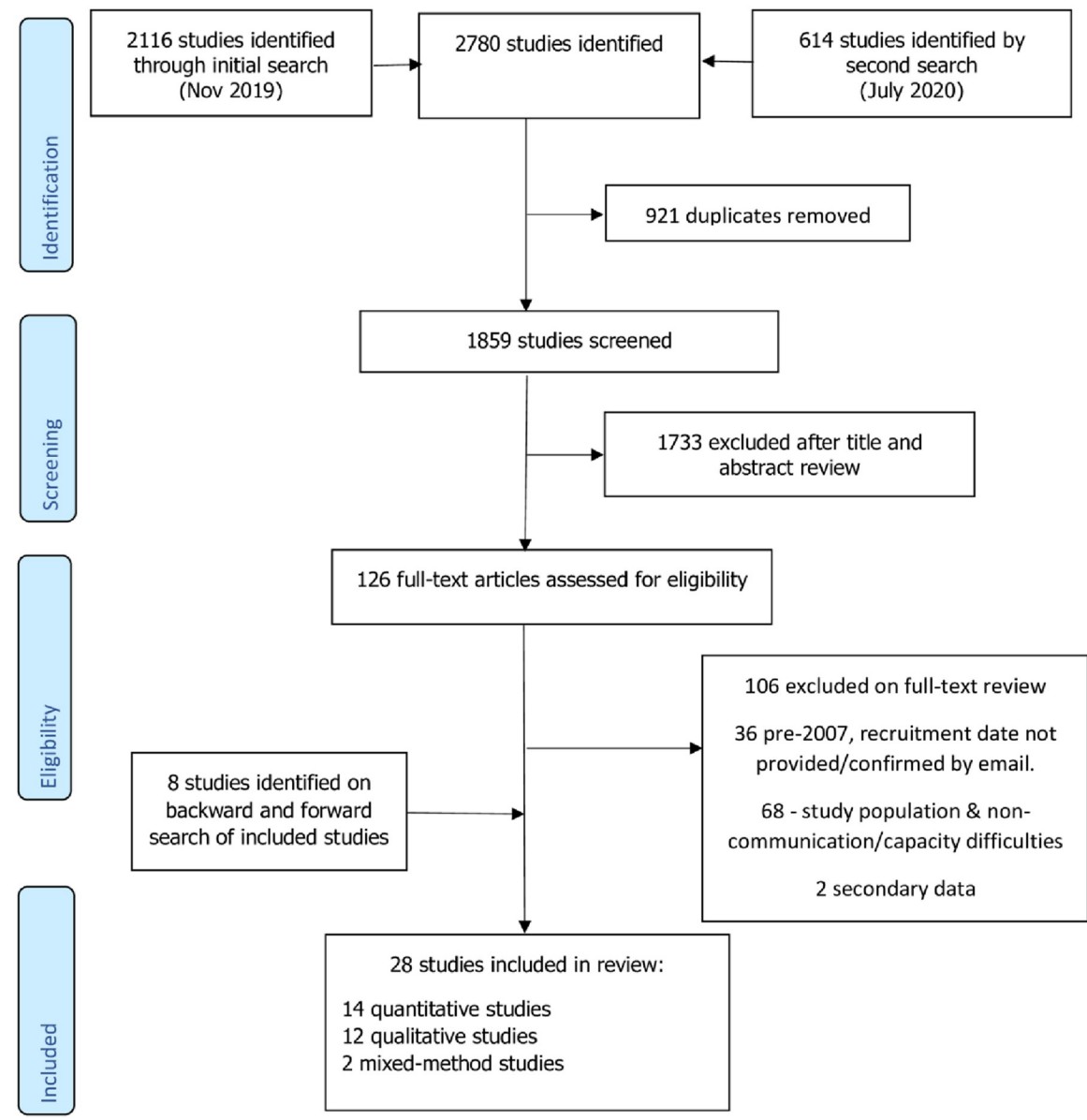

**Fig 1. PRISMA flow diagram of studies included.**

services (n = 13; 46.4%). Others were in receipt of social care services, prisoners, or part of national databases or ongoing studies (n = 15; 53.6%).

## Quality assessment scores

Of the fourteen quantitative studies, twelve (85.7%) were evaluated as high-quality, one (7.1%) as moderate-quality and one (7.1%) as low-quality; all qualitative studies (n = 12, 100%), were evaluated as high-quality and both mixed-methods studies (n = 2, 100%) were evaluated as moderate quality.

All the studies articulated clear research questions and appropriate method to address such questions. Quantitative studies benefitted from the clear description of target population,

use of validated tools and the use of sensitivity analysis and/or adjustments to reduce bias. However, some quantitative studies were weakened by the lack of sample size calculations and the recruitment of only those who had capacity or could speak English language (a potential source of bias). The strength of qualitative studies was based on appropriate methodology, use of triangulation methods, substantiating data with quotes and coherence between data and its interpretation. The quantitative aspect of the two mixed-method studies lacked rigour and clarity. See S4 Table for full details of the quality assessment of each included paper and S5 Table for synopsis of study quality appraisal.

## Identification of participants

In all included studies, participant access was managed through designated gatekeepers, who identified potentially eligible participants. Where specified, the role was variously enacted by clinical practitioners [57–64], other healthcare professionals [65–72], care home managers and staff [73,74], prison staff [75] or support staff [76]. In one study, Hall [74], following a period of acclimatisation in the home, the researcher performed the role of gatekeeper alongside staff and relatives in a residential home for people with dementia.

## Inclusion/Exclusion criteria of participants

Participants deemed to lack capacity were included in 15 studies (54%) based on consultee advice [57–63,68,72,77–82], and excluded from seven studies as part of eligibility criteria (25%) [66,67,69,70,75,76,83]. In one of the studies, potential participants judged not able to consent were not even approached [83]. Of the remaining 6 studies, one made provision for consultee advice but did not use this as all participants were able to give informed consent [84], while the participants in the remaining five studies were able to give informed consent [64,65,71,73,85]. In addition, three studies excluded potential participants based on cognitive-communicative competence for data collection methods [73,83,84], and severe visual and cognitive difficulties [78]. Furthermore, limitations in English as a second language affected exclusions in 3 studies [68,70,75]. The role of personal consultee was fulfilled variously by family members, friends, next of kin, or a close person who knew the participant well [57,58,61,63,64,68,72,74,82,86] while nominated consultees were either paid carers or health-care professionals [59,60,77,81]. Several studies reported checks for verbal and non-verbal signs indicating participant willingness or unwillingness to participate in the research [57,58,67,68,72–75,78,82].

## Study information format

A lack of detail concerning the format of study information was evident in 12 studies (42.9%) [57,58,60–63,66,73,77,79,82,83]. Where detail was provided, the preferred format was text, often combined with verbal explanations [70,72,80,84,85,87]. Wray [76], reported the use of verbal explanation only for those living with aphasia. Eight studies reported adaptations to the participant information sheet in support of communication needs: an 'aphasia friendly' format for people with aphasia post-stroke [78]; 'easy read' versions for people with intellectual disabilities [59,65,71] and ASD/ID [75]; and 'accessible' information for people with intellectual disabilities [67] and dementia [68,69]. One study [59] used graphic images to supplement text. Collaborative development of information sheets by researchers and user group representatives was reported by two studies [68,88] and affected volume of essential information presented [81] and format accessibility [68].

## Further support for decision making process

Supplementary decision-making processes included communicative support from familiar others (e.g. family members, carers, and healthcare professionals) [59]; allowing extra time for participants to process information [65,75]; and providing question and answer opportunities [58,64,68,78,84]. Consideration of setting factors for recruitment activities were also reported: familiar places to minimise any anxiety affecting understanding [58]; and private places to control for distraction [75]. Some studies used a range of information formats and approaches to recruitment. For example, Stoner [69] used a full information sheet, abbreviated, and accessible formats for those living with dementia. While Frighi [59], used a variety of pictures, or 'easy read' materials supplemented by support from familiar others.

## Capacity assessment procedures

Capacity assessment procedures were not reported in detail in many studies. However, authors of 7 studies [57–60,63–65,75] referred to the MCA functional test (MCA 2005), albeit with variously described procedures. Formal assessments were reported for three studies with variable use of closed questions [86]; a checklist of items [65,75]; and standardised questions [85]. Spencer [88], used the MacArthur Competence Assessment Tool for Clinical Research (MacCAT-CR) with people with mental health disorders. It is a semi-structured tool that measures decision-making competence in terms of understanding, appreciation, reasoning and expressing a choice [89]. Informal capacity assessments, appeared to be based on conversations between researcher and prospective participants [72] or on ethnographic observations of the individual's verbal and behavioural responses [67,74] in some studies. Although researchers' judged capacity in most studies, this decision was initially taken by clinicians [60,61,63–65,76,82,90] or other gatekeepers such as care home managers of staff [73,74] or both [66,74]. Individuals deemed to lack capacity were often excluded from research participation without report of a formal assessment [66,67,69–71,75,76].

## Informed consent procedures

Written informed consent was obtained from participants who had capacity to take part in research [57–61,64,65,69,70,72,75–77,80,82,85]. Four studies involving adults with dementia [68,72,80], and intellectual disabilities [67] reported adaptation to the consent process by the use of an enhanced process consent model that monitored ongoing consent through verbal and non-verbal signs, thereby supporting participant autonomy [68,80]. In each case, the researcher maintained a documented 'audit trail' of decisions and actions informed by the gatekeepers and consultees, and the communicative behaviours of participants, as did Hall [80]. Goldsmith [67] assessed consent in adults with intellectual disabilities, by meeting the potential participant with a supporter in attendance and capturing the process on video to document non-verbal cues. This was then checked by the supporter for non-verbal cues to either confirm or deny capacity and a decision that is free from coercion. In addition, one group recruited from a population case register using an 'opt-out consent procedure' and made contact with prospective participants by phone or an 'opt-in consent procedure' where participants contacted the study team directly [79]. A single study [76] used the Consent Support Tool with adults with aphasia post-stroke to determine the requirements for support and the recommended communication strategies.

## Discussion

Our systematic review revealed variable interpretation of the provisions of the MCA (2005) and its accompanying guidance in the CoP. Capacity was included as part of the eligibility

criteria within studies, sometimes as an exclusion criterion. Assessment of capacity is reported inconsistently with some studies adopting formal measures and others making it part of the informed consent procedure. Procedures used for informed and autonomous decision-making appeared to uphold the four defining principles of capacity. Our findings showed that researchers made efforts to maximise individual autonomy through use of various media and tools to support informed consent processes. Beyond seeking a consultee's advice around the inclusion of incapacitous participants, there is limited report of measures to engage such participants in ongoing decisions about participation in research.

The gatekeeper is attributed a pivotal role in gaining access to participants [15,40]. Thus, there is the authority to facilitate or impede recruitment. Furthermore, it is possible that the inclusion of adults with CCD is affected by the gatekeeper's own interpretation of mental capacity for decision-making. Communication difficulties in people post-stroke and memory problems in people with dementia may be mistaken for a lack of capacity by gatekeepers [91]. In one study [68], where all the participants were able to give informed consent, it was asked whether staff acting as gatekeepers avoided those individuals with dementia who had more complex communication needs. This raises questions about the gatekeeper's own agenda and whether support for decision-making gives way to protection. The process whereby gatekeepers decide who to nominate as potential participants lacks clear specification, and may be seen as counter to the MCA [2] requirement for establishing capacity.

A range of strategies were used by researchers to support the accessibility of research information for those with CCD. This is consistent with relevance theory [20], as understanding of research information will be based on the cognitive load of each strategy. The use of accessible information with participants with intellectual disabilities showed compliance with the MCA's second statutory requirement [2,15], reinforced by the Department of Health [18] and the Accessible Information Standards (AIS) [17]. Previous studies have shown that 'aphasia-friendly' study information was preferred by the aphasic participants [92] and led to 11.2% increase in their understanding [93]. This resonates the underlying premise of relevance theory that successful engagement with information requires the least cognitive load [20]. Beyond the use of multiple media to convey information, the support of familiar others and adjusting to individual needs is important [15]. Whilst there was limited report of tailored approaches to supporting CCD, a role for experts-by-experience was exemplified in one study [81], where researcher collaboration with patient group representatives informed the development of study information suitable for those with psychoses. Suitably selected images can support understanding [15]. However, the use of pictures may not be amenable to all participants and interpreted as patronising or misleading [86,92].

Careful consideration and further research are needed to ascertain the best strategies for each group of adults with capacity and communication difficulties.

Recruitment procedures targeting individuals with CCD need to include deliberate measures to achieve the easiest cognitive load possible within the required research framework [20]. Researchers need to be cognisant of the range of strategies and accommodations that can be used to support autonomous decision-making by engaging with the evidence on augmentation and alternative communication methods [19]. This includes the use of picture, simple text, object of reference and supported conversation [17]. In addition, consideration should be given to the individual need of each participant, tailoring accommodations to their preferred way of engaging with researchers [15].

The MCA (2005) recognises people's interest in making decisions as much as possible [2]. An established lack of capacity does not obviate the need to provide opportunities for the participant to express their wishes and feelings. Baumgart proposed the principle of partial participation for individuals with severe developmental disabilities [39]. The concept

embraces the notion of active engagement and advocates 'interdependence' such that individualised adaptations may serve to scaffold participation in ongoing decision-making as far as possible [39].

The lack of detailed description of the MCA's two-staged assessment of capacity process in our findings may be a matter of reporting rather than reality. The use of both formal and informal methods of assessment allowed the inclusion of a range of adults with CCD in research. However, this type of capacity assessment is reported to be less reliable compared with structured assessment in clinical settings [94]. In contrast, our findings showed that ethnographic observations contribute to improved understanding of verbal and non-verbal behaviour and enhance capacity assessment [72,74]. While there is no 'gold standard' method for accessing capacity, the use of an assessment tool was documented in one study [88]. Previous research suggests that the MacCAT-CR tool is adaptable and reliable in those living with dementia and mental health difficulties [89]. There is need for the development and validation of capacity assessment tools in different groups of adults with CCD.

We found that adults who had difficulty communicating and those who were not able to consent to research participation were excluded from research potentially relevant to them. A parallel can be drawn with the clinical trials literature, where similar vulnerable groups were also excluded and therefore remain under-represented in research [95,96]. While eligibility criteria are useful for recruiting participants representative of a target population, exclusions solely based on lack of capacity, without appropriate assessments or adaptations in place are potentially unethical. It is possible that the added demands of consultee procedures and the perceived risks of participation for incapacitous individuals had a negative effect on sample inclusion [95]. This is contrary to Article 12 of the Convention on the Rights of Persons with Disabilities (CRPD: UN 2006) [97] which asserts there should be 'equal recognition before the law'. Their exclusion may skew research sampling and has implications for service provision and policies.

Our findings provide evidence that adults with capacity and communication difficulties can take part in ethically sound research. Adaptations and accommodations are used variously to support both the assessment of capacity and the decision-making process in recruitment of participants, but exclusions still continue on the basis of a lack of capacity.

For the researcher, this means engaging with participants, as well as the gatekeepers and familiar others in their lives who are possible sources of information and support to them. Traditional ways of obtaining informed consent are not appropriate for all, and there is a need to consider the non-traditional ways such as process model of consent. Capacity is relative to a spectrum of decisions. Exercise of capacity can be supported, and its assessment is context- and time-specific. While consultees can facilitate participation in research for those lacking capacity, autonomy through partial participation is possible and to be encouraged. Thus, including people with capacity and communication difficulties in ethically-sound research requires a deliberate approach to devising ways of assessing true capacity and presenting study information.

## Limitations

A possible limitation is that we missed some relevant studies because we excluded publications prior to 2011 in keeping with our focus on the implementation of the MCA. By limiting publication language to only English, we might have missed out on research findings reported in Welsh, the other official language apart from English in Wales. Our search did not yield any study involving adults with acquired brain injury, we have therefore not reported on this population.

## Conclusion

Including adults with CCD in ethically-sound research is a complex proposition demanding deliberate planning of procedures to support autonomous decision-making as far as possible. Furthermore, the complexities of inclusion may cause researchers to err on the side of caution and exclude those deemed/presumed to be incapacitous. There is a need to further investigate the reasoning underpinning researchers' decisions about sample inclusion and the development of research protocols and procedures for participant recruitment. Similarity in the provisions made for those living with dementia, intellectual disability and aphasia implies some common ground for future developments (S1 Fig and S6 Table. Including CCD in research). The use of these strategies may enable researchers to navigate better the recruitment and inclusion adults with CCD in research.

## Supporting information

**S1 Fig. Including CCD in research.**
(TIF)

**S1 Table. Data extraction table.** Showing characteristics and findings of the 28 included papers.
(XLSX)

**S2 Table. PRISMA checklist.** Showing the page numbers on which Preferred Reporting Items for Systematic Reviews and Meta-Analyses (PRISMA) are reported.
(DOCX)

**S3 Table. Summary of the characteristics of included studies with focus on study outcomes.**
(DOCX)

**S4 Table. Quality appraisal of studies using the Mixed Methods Appraisal Tool (MMAT).**
(XLSX)

**S5 Table. Summary table of study synopses (MMAT).**
(DOCX)

**S6 Table. Solutions to CCD recruitment.**
(DOCX)

**S7 Table. Excluded studies.**
(XLSX)

**S1 File. PROSPERO protocol.** Review protocol registered with PROSPERO (International prospective register of systematic reviews).
(PDF)

**S2 File. Search strategy.**
(DOCX)

## Acknowledgments

We would like to thank Marcus Redley for his input into the PROSPERO protocol.

## Author Contributions

**Conceptualization:** Anne Killett, Peter E. Langdon, Rob Heywood, Karen Bunning.

**Data curation:** Oluseyi Florence Jimoh, Hayley Ryan, Karen Bunning.

**Formal analysis:** Oluseyi Florence Jimoh.

**Funding acquisition:** Karen Bunning.

**Methodology:** Oluseyi Florence Jimoh, Hayley Ryan, Anne Killett, Ciara Shiggins, Peter E. Langdon, Rob Heywood, Karen Bunning.

**Project administration:** Karen Bunning.

**Writing – original draft:** Oluseyi Florence Jimoh.

**Writing – review & editing:** Oluseyi Florence Jimoh, Hayley Ryan, Anne Killett, Ciara Shiggins, Peter E. Langdon, Rob Heywood, Karen Bunning.

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
