## [Decision Letter · Decision Letter 0]

9 Jun 2021

PONE-D-21-15341

Applying the provisions for research under the Mental Capacity Act (2005) in England and Wales to adults with capacity and communication difficulties: a systematic review and narrative synthesis

PLOS ONE

Dear,

Thank you for submitting your manuscript to PLOS ONE. After careful consideration, we feel that it has merit but does not fully meet PLOS ONE’s publication criteria as it currently stands. Therefore, we invite you to submit a revised version of the manuscript that addresses the points raised during the review process.

Kindly addressed all comments given by reviewer.

We look forward to receiving your revised manuscript.

Kind regards,

Muhammad Shahzad Aslam, Ph.D.,M.Phil., Pharm-D

Academic Editor

PLOS ONE

Journal Requirements:

2. PLOS ONE does not copy edit accepted manuscripts (https://journals.plos.org/plosone/s/criteria-for-publication#loc-5). To that effect, please ensure that your submission is free of typos and grammatical errors.

Reviewers' comments:

Reviewer's Responses to Questions

**Comments to the Author**

1. Is the manuscript technically sound, and do the data support the conclusions?

Reviewer #1: Yes

Reviewer #2: Partly

2. Has the statistical analysis been performed appropriately and rigorously? 

Reviewer #1: Yes

Reviewer #2: N/A

3. Have the authors made all data underlying the findings in their manuscript fully available?

Reviewer #1: Yes

Reviewer #2: Yes

4. Is the manuscript presented in an intelligible fashion and written in standard English?

Reviewer #1: Yes

Reviewer #2: Yes

5. Review Comments to the Author

Reviewer #1: General Comments

I think this is a very systematic review, it is important to evaluate how consent is obtained in vulnerable populations. It is also clear that participants who could be able to take part in research should not be excluded. Some areas of the manuscript needed some further clarification/justification in order to improve it.

Specific comments

Consider revising the title to

“A systematic review and narrative synthesis of the provisions used to recruit adults with capacity and communication difficulties under the mental capacity act (2005) in England and Wales”

Consider revising the first sentence in the abstract to this:

“The Mental Capacity Act (2005) as well as the Code of Practise (2007), governs whether adults with capacity and communication difficulties are able to participate in research in England and Wales.”

Last sentence of conclusion in the abstract needs to clarify why it is important that they are included – suggest adding a sentence to justify that inclusion could assist care decisions

The range of disorders/disease is wide – why did you select these in particular, this needs to be justified?

Why were studies involving participants aged above 16 years and not 18? This could be expanded when discussing the inclusion criteria of the studies.

Are the rules different in Scotland and Ireland? If so, this needs to be clarified as to why you only focused on studies conducted in England and Wales.

Is 28 studies sufficient to conduct a systematic review? How many papers were given an MMAT score of 3 or less? Does excluding the lower scoring publications change the narrative?

Since you were looking at studies carried out in Wales – why was the inclusion criteria English only? Did some of the other studies mention that Welsh wasn’t the other languages spoken?

Fig 1 says 29 studies included but 28 in the abstract?

In discussion, clarify which strategy or strategies would best be used to allow people with reduced capacity to consent. This is a little implicit.

Reviewer #2: Dear respectful editorial manager,

Thank you for inviting me as reviewer for manuscript entitled “Applying the provisions for research under the Mental Capacity Act (2005) in England and Wales to adults with capacity and communication difficulties: a systematic review and narrative synthesis”

My observations and minor concerns are:

1. In the abstract total studies included is 28 but in flow diagram 29

2. P3/L48 CCD but P 4/L92 CDD in S1 ICC. Please be consistent.

3. Only 2/28 studies included are mix-method, why MMAT as the main quality appraisal?

4. Page 17: line 226: Capacity assessment procedures were typically not reported in detail. Please change this style of writing.

My strongly suggest for major improvement .

1.The introduction was not well articulated, the research gap and significant of the study are not clear.

2. Theoretical part for on CCD is missing and little available. We recommend authors to add some theoretical ideas at introduction and also discussion.

3. Since the majority (85.7%) of the studies included were quantitative, we suggest authors to use a PICO frame.

References

We scan all references. Some of them are incomplete.

Reference 3/ 4 / 53 /78

Therefore, I recommend this paper for publication with major revision.

6. PLOS authors have the option to publish the peer review history of their article (what does this mean?). If published, this will include your full peer review and any attached files.

Reviewer #1: No

Reviewer #2: **Yes: **Siti Aishah Hassan

---

## [Author Response · Author response to Decision Letter 0]

26 Jun 2021

Thank you for recommending our paper for publication with major revision.

We have tried our best to address all the issues raised. Please see our response to comments below. 

Reviewer #1

Reviewer 1: I think this is a very systematic review, it is important to evaluate how consent is obtained in vulnerable populations. It is also clear that participants who could be able to take part in research should not be excluded. Some areas of the manuscript needed some further clarification/justification in order to improve it.

Authors’ response: Thank you for your comments on our review. We have tried out best to address the issue raised. 

Specific comments

1. Consider revising the title to “A systematic review and narrative synthesis of the provisions used to recruit adults with capacity and communication difficulties under the mental capacity act (2005) in England and Wales”

Authors’ response: Thank you for your suggestion which we have adopted. We have broken it up with a colon to make it easier to follow. The title now reads:

 “A systematic review and narrative synthesis of the research provisions under the Mental Capacity Act (2005) in England and Wales: Recruitment of adults with capacity and communication difficulties” (Page 1, lines 1 & 2).

2. Consider revising the first sentence in the abstract to this: “The Mental Capacity Act (2005) as well as the Code of Practise (2007), governs whether adults with capacity and communication difficulties are able to participate in research in England and Wales.”

Authors’ response: Thank you for suggestion. This has now been revised and now reads:

“The Mental Capacity Act (MCA, 2005) and its accompanying Code of Practise (2007), govern research participation for adults with capacity and communication difficulties in England and Wales.” (Page 2, Lines 21 & 22).

3. Last sentence of conclusion in the abstract needs to clarify why it is important that they are included – suggest adding a sentence to justify that inclusion could assist care decisions.

Authors’ response: Thank you for your comment. 

It is beyond the scope of this review to make suggestions about care decisions as it is aimed at research procedures. We have added an explanation to justify their inclusion in research. Details below: 

“If health and social interventions are to be effective for this population, they need to be included in primary research. The use of strategic adaptations and accommodations during the recruitment process, may serve to support their inclusion.” (Page 2, Lines 42 - 44).

4. The range of disorders/disease is wide – why did you select these in particular; this needs to be justified?

Authors’ response: Thank you for your comment. 

We agree that the range of capacity and communication affecting disorder is wide. We were not interested in the whole range but have chosen to focus on six main capacity affecting conditions where we have associated communication difficulties. 

We have rephrased our introductory section to reflect this (Page 3, Lines 47 - 54). 

5. Why were studies involving participants aged above 16 years and not 18? This could be expanded when discussing the inclusion criteria of the studies.

Authors’ response: Thank you for your comment.

We included studies with participants aged 16 years and above because this is the age at which the MCA applies. The inclusion criteria have been altered to:

Participants aged 16+ years (the age at which the MCA applies), with communication and/or capacity difficulties (e.g. associated with autism; stroke; mental health; dementia; acquired brain injury; and intellectual disabilities (Page 7, Table 1).

6. Are the rules different in Scotland and Ireland? If so, this needs to be clarified as to why you only focused on studies conducted in England and Wales.

Authors’ response: Thank you for your comment.

The rules are slightly different in Scotland and Ireland. We have added a few sentences to the manuscript to acknowledge that there are rules in others, but the focus of our review is England and Wales. The researchers are based in England and have focused on the law guiding their own practice. Please see the addition below: 

“There are separate provisions for research (CoP: Chapter 11). Different legislation is provided in other countries of the UK: the Adults with Incapacity (Scotland) Act 2000 (AWIA); the Mental Capacity Act (Northern Ireland) (2016). In Ireland, it is the Assisted Decision Making (Capacity) Act 2015. However, the current review pertains to the Mental Capacity Act (2005) in England and Wales.”. (Page 3, Lines 62-65).

7. Is 28 studies sufficient to conduct a systematic review? How many papers were given an MMAT score of 3 or less? Does excluding the lower scoring publications change the narrative?

Authors’ response: Thank you for your observation. 

Yes, 28 studies are enough to conduct a systematic review. 

The 28 studies are the outcome of the systematic search of the literature. There are no present number of studies that can be included. It all depends on how many studies fit the inclusion criteria. 

Of all 28 studies, 24 were high quality, two were moderate (score 3), one scored 2. Excluding the lower scoring publications (n=3), does not change the narrative of the review.

8. Since you were looking at studies carried out in Wales – why was the inclusion criteria English only? Did some of the other studies mention that Welsh wasn’t the other languages spoken?

Authors’ response: Thank you for your observation.

We considered that since our focus was on the implementation of the Mental Capacity Act in England and Wales, most articles will be reported in English Language. Moreover, most articles written in Welsh will usually have an English version. In addition, we did not have a Welsh speaker on the research team which limited our ability to take on any Welsh article which has no translation. It is possible that we missed out on some of those. This has now been acknowledged as a weakness. See below:

“By limiting publication language to only English, we might have missed out on research findings reported in Welsh, the other official language apart from English in Wales.” (Page 17/18, Lines 373-375).

9. Fig 1 says 29 studies included but 28 in the abstract?

Authors’ response: Thank you for the observation. This is a typo and has now been corrected. (see Figure 1, page 9).

10. In discussion, clarify which strategy or strategies would best be used to allow people with reduced capacity to consent. This is a little implicit.

Authors’ response: Thank you for your suggestion. 

The following has now been added to the discussion section:

“Recruitment procedures targeting individuals with CCD need to include deliberate measures to achieve the easiest cognitive load possible within the required research framework (1). Researchers need to be cognisant of the range of strategies and accommodations that can be used to support autonomous decision-making by engaging with the evidence on augmentation and alternative communication methods (2). This includes the use of picture, simple text, object of reference and supported conversation (3). In addition, consideration should be given to the individual need of each participant, tailoring accommodations to their preferred way of engaging with researchers (4)”. (Page 15, Lines 324-330).

Reviewer #2

1. In the abstract total studies included is 28 but in flow diagram 29

Authors’ response: Thank you for spotting that. Apologies, it is a typo. There are 28 studies and has now been corrected. See figure 1: PRISMA flow diagram of studies included. (Page 9, Figure 1).

2. P3/L48 CCD but P 4/L92 CDD in S1 ICC. Please be consistent.

Authors’ response: Thank you for noticing the discrepancy. 

The correct acronym is CCD. It stands for Capacity and Communication difficulties. CDD in S1 ICC is in reference to our published protocol on Prospero (A supplementary document for the current manuscript), it was not used in the current manuscript. 

In the earlier stages of our project, we used the term “adults with impairments of capacity and/or communication (ICC)”, as reflected in our protocol. This was later refined to “adults with capacity and communication difficulties” to make the term easier for our stakeholders. 

We have now included a sentence in the methods section of our review to reflect the change of term. The sentence reads:

“In the protocol, we used the term “adults with impairments of capacity and/or communication (ICC)”. This has been refined and modified through our interactions with our stakeholders to “adults with capacity and communication difficulties”. (See Page 5/6, Lines: 124-126).

3. Only 2/28 studies included are mix-method, why MMAT as the main quality appraisal?

Authors’ response: Thank you for your observation.

We used the Mixed Methods Appraisal Tool (MMAT) for consistency in assessing the quality of all included studies. 

The following has now been added to the ‘Data extraction and quality assessment’ section of the “Methods” to justify our use of their tool:

“The Mixed Methods Appraisal Tool (MMAT) (5), for concurrent critical appraisal of quantitative, qualitative and mixed-methods primary research was applied (6). The MMAT has established content validity, it has been piloted across all methodologies; quantitative, qualitative and mixed methods research designs (6,7). Compared with other tools, the MMAT includes specific criteria for appraising mixed methods studies. While critical appraisal tools are more widely available for quantitative and qualitative research, there has not been consensus on quality criteria for mixed methods research (8)”. (Page 8, lines: 159-164).

4. Page 17: line 226: Capacity assessment procedures were typically not reported in detail. Please change this style of writing.

Authors’ response: Thank you for your comment. We have modified this to:

“Capacity assessment procedures were not reported in detail in many studies.” (P12, Line 259).

Further comments (Academic editor)

1. The introduction was not well articulated, the research gap and significant of the study are not clear. 

Authors’ response: Thank you for your comment.

The literature review has now been updated, research gap identified, and the significance improved. Please see the introductory section. 

2. Theoretical part for on CCD is missing and little available. We recommend authors to add some theoretical ideas at introduction and also discussion.

Authors’ response: Thank you for your suggestion.

The introduction and discussion sections have been updated with some theoretical ideas. 

Introduction:

 “For the purposes of research, there is the presumption of capacity unless there is a reason to believe that a person lacks capacity (CoP 2007). Before deciding that someone lacks capacity, the CoP (2007) recommends the provision of relevant information, communicated in the most appropriate way (4). Whilst practical details are not given, there is general encouragement for presenting project information to suit the processing capabilities of potential participants. For example, support for the person’s understanding of what research participation entails might include: information sheets rendered in simple language with or without pictorial support; a simulated data collection procedure shown on video; questions and answer opportunities in conversations about a project; and use of manual sign and gesture to augment meanings (2–4,9). Relevance theory (1) argues that people find it easier to engage with and understand information that is most relevant to them and requires the least cognitive effort. The form of the message interacts with the person’s cognitive abilities, prior experience and underlying knowledge. On this latter point, the person’s familiarity with the subject matter contributes to their perception of possible cognitive gain, which in turn optimises the potential relevance of information to them (1). This asserts the importance of addressing the information-processing needs of the target population for successful recruitment to studies, particularly where CCDs are present.” (P3/4, Lines 70-84).

Discussion 

A range of strategies was used by researchers to support the accessibility of research information for those with CCD. “This is consistent with relevance theory (1), as understanding of research information will be based on the cognitive load of each strategy”.

Previous studies have shown that ‘aphasia-friendly’ study information was preferred by the aphasic participants (10) and led to 11.2% increase in their understanding (11). “This resonates the underlying premise of relevance theory that successful engagement with information requires the least cognitive load (1).”

3. Since the majority (85.7%) of the studies included were quantitative, we suggest authors to use a PICO frame.

Authors’ response: Thank you for your comment.

The PICO framework is better suited for therapy questions. We have used it as much as possible in Table 1, to describe the eligibility criteria for included studies. please see Table 1 for details (page 7).

4. References We scan all references. Some of them are incomplete. Reference 3/ 4 / 53 /78 Therefore, I recommend this paper for publication with major revision.

Authors’ response: Thank you for your observation. The reference identified have now been revised.

We hope we have been able to respond to all your queries appropriately.

Authors: Oluseyi F. Jimoh and colleagues

References

1. Sperber D, Wilson D. Relevance: Communication and Cognition. 2nd Editio. Oxford: Blackwell Publishing Inc; 1986. 1995 p. 

2. Beukelman D, Light J. Augmentative & Alternative Communication: Supporting Children and Adults with Complex Communication Needs. 5th editio. London, UK: Blackwell Publishing Inc; 2020. 

3. MENCAP. Accessible Information Standard [Internet]. Available from: https://www.mencap.org.uk/accessible-information-standard

4. Department for Constitutional Affairs. Mental Capacity Act 2005 Code of Practice [Internet]. London: The Stationary Office. 2007. p. 1–301. Available from: https://assets.publishing.service.gov.uk/government/uploads/system/uploads/attachment_data/file/497253/Mental-capacity-act-code-of-practice.pdf

5. Hong Q, Fàbregues S, Bartlett G, Boardman F, Cargo M, Dagenais P, et al. (10 more authors) (2018) The Mixed Methods Appraisal Tool (MMAT) version 2018 for information professionals and researchers. Educ Inf. 2018;34(4):285–91. 

6. Pace R, Pluye P, Bartlett G, Macaulay A, Salsberg J, Jagosh J, et al. Testing the reliability and efficiency of the pilot Mixed Methods Appraisal Tool (MMAT) for systematic mixed studies review. Int J Nurs Stud J. 2012;49(1):Epub 2011 Aug 10. PMID: 21835406. 

7. Souto R, Khanassov V, Hong Q, Bush P, Vedel I, Pluye P. Systematic mixed studies reviews: updating results on the reliability and efficiency of the mixed methods appraisal tool. Int J Nurs Stud. 2015;52:500–1. 

8. O’Cathain A, Murphy E, Nicholl J. The quality of mixed methods studies in health services research. J Health Serv Res Policy. 2008;92-8.(2):92–8. 

9. Department of Health. Making written information easier to understand for people with learning disabilities: Guidance for people who commission or produce Easy Read information – Revised Edition 2010. Dep Heal. 2010;1–40. 

10. Taylor J, DeMers S, Vig E, Borson S. The disappearing subject: exclusion of people with cognitive impairment and dementia from geriatrics research. J Am Geriatr Soc. 2012;60:413–9. 

11. Rose TA, Worrall LE, Hickson LM, Hoffmann TC. Aphasia friendly written health information: Content and design characteristics. Int J Speech Lang Pathol. 2011;13(4):335–47.

---

## [Decision Letter · Decision Letter 1]

13 Aug 2021

A systematic review and narrative synthesis of the research provisions under the Mental Capacity Act (2005) in England and Wales: Recruitment of adults with capacity and communication difficulties

PONE-D-21-15341R1

Dear,

We’re pleased to inform you that your manuscript has been judged scientifically suitable for publication and will be formally accepted for publication once it meets all outstanding technical requirements.

Kind regards,

Muhammad Shahzad Aslam, Ph.D.,M.Phil., Pharm-D

Academic Editor

PLOS ONE

Additional Editor Comments (optional):

Reviewers' comments:

Reviewer's Responses to Questions

**Comments to the Author**

1. If the authors have adequately addressed your comments raised in a previous round of review and you feel that this manuscript is now acceptable for publication, you may indicate that here to bypass the “Comments to the Author” section, enter your conflict of interest statement in the “Confidential to Editor” section, and submit your "Accept" recommendation.

Reviewer #1: All comments have been addressed

Reviewer #2: All comments have been addressed

2. Is the manuscript technically sound, and do the data support the conclusions?

Reviewer #1: (No Response)

Reviewer #2: Yes

3. Has the statistical analysis been performed appropriately and rigorously? 

Reviewer #1: (No Response)

Reviewer #2: N/A

4. Have the authors made all data underlying the findings in their manuscript fully available?

Reviewer #1: (No Response)

Reviewer #2: Yes

5. Is the manuscript presented in an intelligible fashion and written in standard English?

Reviewer #1: (No Response)

Reviewer #2: Yes

6. Review Comments to the Author

Reviewer #1: (No Response)

Reviewer #2: Good enough

7. PLOS authors have the option to publish the peer review history of their article (what does this mean?). If published, this will include your full peer review and any attached files.

Reviewer #1: No

Reviewer #2: **Yes: **Siti Aishah Hassan

---

## [Editor Report · Acceptance letter]

23 Aug 2021

PONE-D-21-15341R1 

A systematic review and narrative synthesis of the research provisions under the Mental Capacity Act (2005) in England and Wales:  Recruitment of adults with capacity and communication difficulties 

Dear Dr. Jimoh:

I'm pleased to inform you that your manuscript has been deemed suitable for publication in PLOS ONE. Congratulations! Your manuscript is now with our production department. 

Kind regards, 

on behalf of

Dr. Muhammad Shahzad Aslam 

Academic Editor

PLOS ONE